# Dichloroacetic Acid Enhances Photodynamic Therapy-Induced Regulated Cell Death in PANC-1 Pancreatic Cancer Cell Line

**DOI:** 10.3390/ijms262211031

**Published:** 2025-11-14

**Authors:** Adeolu S. Oluremi, Krishnaswamy Kannan, Nawab Ali

**Affiliations:** 1Department of Biology, Donaghey College of Science, Technology, Engineering and Mathematics, University of Arkansas at Little Rock, Little Rock, AR 72204, USA; asoluremi@ualr.edu; 2Biomolecular Integrations, Little Rock, AR 72205, USA

**Keywords:** pancreatic cancer, photodynamic therapy, dichloroacetic acid, 5-aminolevulinic acid, oxidative stress, reactive oxygen species, apoptosis, immunogenic cell death, ferroptosis

## Abstract

Pancreatic ductal adenocarcinoma (PDAC) is a highly lethal malignancy characterized by late-stage diagnosis, dense stromal barriers, and resistance to conventional therapies. The tumor microenvironment (TME), marked by hypoxia, aberrant vasculature, and metabolic reprogramming, supports tumor persistence and immune evasion. Targeting metabolic and oxidative vulnerabilities in the TME offers a promising strategy to improve treatment outcomes. This study evaluated the combined effects of photodynamic therapy (PDT) using 5-aminolevulinic acid (5-ALA), a precursor to the natural photosensitizer protoporphyrin IX (PpIX), and dichloroacetic acid (DCA), a mitochondrial function modulator, in the *KRAS*-mutated PANC-1 pancreatic cancer cell line. The combination of 5-ALA–PDT and DCA significantly reduced cell viability compared with either treatment alone. Mechanistic analyses revealed activation of multiple regulated cell death pathways, including mitochondria-mediated apoptosis, immunogenic cell death (ICD), and ferroptosis. This was evidenced by increased reactive oxygen species (ROS), loss of mitochondrial membrane potential (ΔΨm), release of danger-associated molecular patterns (DAMPs) such as ATP, and lipid peroxidation. DCA amplified PDT-induced oxidative stress, overcoming redox defenses and enhancing ferroptotic and immunogenic responses. These findings suggest that combining DCA with PDT enhances multimodal cell death in PDAC, providing a rationale for further in vivo studies to validate this redox–metabolic approach to treating chemoresistant pancreatic tumors.

## 1. Introduction

Pancreatic ductal adenocarcinoma (PDAC) remains one of the most aggressive and lethal human malignancies, with a five-year survival rate below 10% and a median survival of 5–8 months [1]. Its dismal prognosis arises from several converging factors: delayed clinical presentation due to the pancreas’s deep anatomical location, early metastatic spread, profound chemoresistance, and a highly immunosuppressive tumor microenvironment (TME) [2]. In 2023, the American Cancer Society reported approximately 64,050 new pancreatic cancer cases and 50,550 deaths in the United States, with similar rates across sexes [1]. However, these regimens are limited by systemic toxicity, low tumor penetrance, and the development of intrinsic and acquired resistance [3,4,5]. Dense stromal fibrosis, hypoxia, aberrant angiogenesis, and metabolic plasticity further compound therapeutic failure in PDAC [6,7].

Recent studies highlight metabolic plasticity and resistance to regulated cell death (RCD) as hallmarks of PDAC’s tenacity [3,7,8]. While apoptosis, a programmed cell death pathway, is often thwarted by *TP53* mutations and anti-apoptotic proteins like Bcl-2 [5], emerging research points to ferroptosis—an iron-dependent, oxidative cell death process—as a promising target. Ferroptosis, driven by lipid peroxidation and reactive oxygen species (ROS), exploits PDAC’s high basal ROS levels, particularly in *KRAS*-mutated cells [9,10]. Additionally, immunogenic cell death (ICD) offers a unique opportunity to stimulate anti-tumor immunity through damage-associated molecular patterns (DAMPs) like ATP and calreticulin [11]. However, single-agent strategies targeting these pathways often falter due to compensatory mechanisms in PDAC’s heterogeneous TME.

Despite the therapeutic promise of ferroptosis, attempts to induce it as a single-agent strategy in PDAC have yielded limited efficacy. This is largely due to compensatory antioxidant networks and the heterogeneous ferroptotic threshold across tumor subpopulations. Consequently, combinatorial approaches that amplify oxidative stress or disrupt metabolic resilience are under active investigation. One such approach involves photodynamic therapy (PDT), a minimally invasive modality that induces spatially confined oxidative damage. PDT employs photosensitizers (PSs) that, upon excitation by specific wavelengths of light, transfer energy to molecular oxygen, thereby generating highly reactive species such as singlet oxygen and other ROS. 5-Aminolevulinic acid (5-ALA) is a well-established prodrug photosensitizer that is metabolized intracellularly into natural protoporphyrin IX within target cells.

While PDAC cells are not inherently resistant to PDT under standard culture conditions, clinical application faces significant hurdles [3]. The dense tumor microenvironment (TME) of PDAC, coupled with the deep anatomical location of the pancreas, presents considerable practical and physiological limitations to the effectiveness of conventional PDT, ultimately leading to substantial treatment challenges [3,12]. Under in vitro conditions, ROS generation during PDT overwhelms cellular antioxidant defenses, leading to lipid peroxidation and membrane destabilization, the hallmarks of ferroptosis [13,14]. Complementing PDT’s ROS-generating capacity, dichloroacetic acid (DCA) targets cancer metabolism by inhibiting pyruvate dehydrogenase kinase (PDK), thereby redirecting pyruvate into the mitochondria to enhance oxidative phosphorylation [15]. This metabolic shift increases mitochondrial ROS production while reducing intracellular GSH levels, sensitizing cells to oxidative stress and potentially enhancing ferroptotic and apoptotic signaling [16,17]. Thus, mitochondria play a crucial role in PDAC’s aggressiveness and therapeutic resistance due to bioenergetic and metabolic adaptations within the TME. Targeting these underlying mechanisms has the potential to develop novel therapies to treat and overcome resistance [18].

The PANC-1 cell line represents a widely accepted in vitro model for PDAC due to its genotypic and phenotypic fidelity to clinical tumors [19]. These attributes not only contribute to metastatic potential and chemoresistance but also confer susceptibility to ferroptosis via increased oxidative stress and compromised redox buffering capacity.

Beyond apoptosis and ferroptosis, immunogenic cell death (ICD) has garnered significant attention as a mechanism of therapy-induced tumor control. ICD is a form of regulated cell death that stimulates the adaptive immune response via the release of damage-associated molecular patterns (DAMPs), including ATP, calreticulin exposure, and HMGB1 secretion. These immunogenic signals act on dendritic cells and macrophages, enhancing antigen presentation and effector T-cell activation [11]. PDT is among the few therapies with clinically demonstrated ICD-inducing capacity, largely attributable to its spatiotemporal control of ROS generation and tumor-selective cytotoxicity [20].

In this study, we hypothesize that the combined treatment of PDT and DCA will enhance cell death compared with either treatment alone by triggering multiple cell death mechanisms involving redox and metabolic stress. Therefore, we systematically investigated the combined effect of 5-ALA-based PDT and DCA on inducing multiple regulated cell death pathways, specifically apoptosis, ferroptosis, and immunogenic cell death, in PANC-1 cells. Our experimental approach included assessments of ΔΨm, caspase activity, lipid peroxidation, and DAMP release to understand the underlying regulated cell death mechanisms. Our findings suggest that this therapeutic combination induces activation of apoptotic, immunogenic, and ferroptotic pathways. This study not only establishes the feasibility of targeting PDAC through a redox–metabolic axis but also offers a compelling rationale for further translational studies using other cell lines (e.g., Mia PaCa-2 or AsPc-1), patient-derived organoids, and immunocompetent in vivo models. By integrating redox biology, metabolic modulation, and immunogenic death signaling, this strategy may help overcome the therapeutic inertia that defines pancreatic cancer.

## 2. Results

To elucidate the therapeutic potential of photodynamic therapy (PDT) in combination with dichloroacetate (DCA), for pancreatic cancer, this study employed the PANC-1 cell line as an in vitro model to systematically evaluate its capacity to induce regulated cell death (RCD) and inhibit proliferation. Results are presented in a progressive order, beginning with the effects of single-agent treatment (dichloroacetate (DCA), photosensitizer (5-ALA), and laser light irradiation) to optimize doses. This was followed by combination of dual-agent treatments, culminating in the combined effects of PDT and DCA. We first assessed core outcomes, including cell viability, proliferation, and RCD markers, with particular emphasis on mitochondria-driven mechanisms, including mitochondrial membrane permeabilization, ROS production, ATP release, and lipid peroxidation, which underpin enhanced cytotoxicity. Our results suggest the involvement of specific RCD pathways, progressing from apoptosis to immunogenic cell death signatures and finally ferroptosis engagement.

### 2.1. DCA-Induced Changes in Cell Viability and Proliferation in PANC-1 Cells

To identify a sub-toxic dose of DCA to be used safely in subsequent combination treatment studies, we first evaluated its cytotoxic effects on PANC-1 cell viability and proliferation. Treatment with DCA for 24 h reduced cell viability in a concentration-dependent manner, as measured by the MTT assay, with an IC_50_ of 39 mM (Figure 1A). This cytotoxicity was corroborated by the LDH release assay, which also showed a dose-dependent increase in extracellular LDH (Figure 1B). Further, DCA suppressed cell proliferation in a concentration- and time-dependent fashion (Figure 1C). For instance, at 20 mM DCA, proliferation declined to 50% of control levels after 48 h and further to 25% after 96 h. Compared with the control, a significant decline in cell proliferation was evident across most DCA concentrations at 96 h, whereas shorter time periods (24–48 h) required doses >30 mM for comparable effects. Based on these findings, we selected 30 mM DCA for 24 h treatment as a non-cytotoxic regimen for combination therapy experiments.

### 2.2. DCA-Induced Cytotoxicity Is Mediated by Apoptosis

To determine whether the cytotoxicity observed with DCA (Figure 1) was mediated by apoptosis, we assessed early and late apoptotic events in PANC-1 cells using flow cytometry with an Annexin V-FITC/propidium iodide (PI) staining kit. As shown in Figure 2A, DCA treatment induced apoptosis in a concentration-dependent manner, with early and late stages evident in the lower- and upper-right quadrants, respectively. Low doses (10–30 mM) elicited minimal apoptosis, whereas higher concentrations (>30 mM) triggered a marked, dose-dependent increase primarily in late apoptosis. Quantitative analysis confirmed this trend, with total apoptotic cells increasing from <10% at 10 mM to >40% at 70 mM DCA (Figure 2B). These findings indicate that DCA-mediated cytotoxicity is predominantly driven by induction of apoptosis.

### 2.3. DCA Induces Intrinsic Apoptosis via the Mitochondrial Pathway

To determine whether DCA-induced apoptosis engages the intrinsic mitochondrial pathway, we evaluated the activation of caspases-3 and -7, canonical effectors of this cascade. As described in the Materials and Methods Section, flow cytometry analysis was performed using the CellEvent™ Caspase-3/7 Green Detection Reagent, a cell-permeant substrate comprising a DEVD peptide sequence conjugated to a nucleic acid-binding fluorogenic dye. Upon caspase-3/-7-mediated cleavage of the DEVD motif during apoptosis, the released dye binds DNA, yielding a bright signal with excitation/emission maxima at 502/523 nm. Co-staining with SYTOX™ AADvanced™ Dead Cell Stain enabled discrimination of early/late apoptotic cells (caspase-positive, SYTOX-negative) from viable (double negative) and necrotic (double positive) populations. Representative flow cytometry plots (Figure 3A) revealed a dose-dependent increase in caspase-3/-7 activation, marked by progressive shifts toward the early (lower-right quadrant) and late (upper-right quadrant) apoptotic fractions. Quantitative analysis of pooled data (Figure 3B) confirmed that total apoptosis remained modest at ~15% with 40 mM DCA but escalated markedly at higher concentrations (e.g., >60% at 70 mM). These findings indicate that DCA elicits dose-dependent apoptosis through activation of the intrinsic mitochondrial pathway, highlighting its potential as a modulator of PANC-1 cell fate.

### 2.4. Photodynamic Therapy Synergizes DCA-Induced Cytotoxicity

Photodynamic therapy (PDT) in this study entailed incubating cells with 5-aminolevulinic acid (5-ALA), a naturally occurring photosensitizer, for 4 h, followed by irradiation with a 635 nm infrared laser. To establish baseline effects, we first assessed single-agent treatments of 5-ALA, laser irradiation, and their combination, with or without DCA, on PANC-1 cell viability. 5-ALA alone at concentrations of 0.5–2.0 mM exerted no significant impact on cell viability (Figure 4A). Likewise, laser irradiation alone at fluences of 9–108 J/cm^2^ proved non-toxic (Figure 4B). However, preincubation with 1.0 mM 5-ALA followed by laser exposure significantly reduced cell viability at higher fluences (27–108 J/cm^2^; Figure 4C). In contrast, a sublethal dose of DCA (30 mM), when combined with laser irradiation in the absence of the photosensitizer, did not alter cell viability significantly. These findings confirm that PDT elicits cytotoxicity independently at elevated laser doses, setting the stage for synergistic interactions with DCA.

We have previously established that DCA potentiates PDT-induced apoptosis in breast cancer (MCF-7) cells [21]. However, to our knowledge, there is currently no research available on the combined effect of PDT and DCA on pancreatic cancer cells, but their individual effects suggest a potential synergistic strategy. The overall objective of our experimental design was to enhance the effectiveness of PDT in combination with DCA. Such a strategy can be applied to the in vivo conditions where PDT has limited penetration because of the dense TME.

In this study, we examined whether DCA synergizes with the PDT effect in PANC-1 cells. Figure 5 shows cell viability with PDT using various doses of 5-ALA followed by 30 mM DCA treatment. The non-toxic dose of DCA (30 mM) significantly decreased PANC-1 cell viability only when subjected to PDT treatment first. Cell viability decreased significantly with the increase in doses of 5-ALA and laser irradiation. These results suggest that DCA significantly decreased cell viability following PDT compared with their respective controls.

### 2.5. Underlying Mechanisms of PDT-DCA-Induced Cell Death

Our findings indicate that DCA augments PDT-mediated cytotoxicity in PANC-1 cells. To understand the underlying mechanisms, we evaluated mitochondrial integrity, reactive oxygen species (ROS) production, and ATP release as markers of immunogenic cell death (ICD). As shown in Figure 6A, combination of PDT with 30 mM DCA (PDT-DCA) induced profound mitochondrial damage relative to single-agent or control treatments. This regimen also elicited robust ROS generation, a hallmark of oxidative stress (Figure 6B). Moreover, PDT-DCA triggered substantial ATP efflux, signifying ICD induction (Figure 6C). Collectively, these data demonstrate that PDT-DCA elicits cellular stress culminating in mitochondrial dysfunction, ROS accumulation, and immunogenic demise, thereby amplifying anti-tumor effects.

### 2.6. PDT-DCA Induces Mitochondria-Mediated Intrinsic Apoptosis Pathway

To determine whether PDT-DCA-induced mitochondrial damage, ROS production, and ATP release drive intrinsic apoptosis, we employed flow cytometry analysis using Annexin V-FITC/propidium iodide (PI) staining, as outlined in the Materials and Methods Section. Analysis by flow cytometry revealed that PDT + DCA resulted in a significantly higher population of apoptotic cells (Figure 7A). Post-PDT at 108 J/cm^2^, DCA-mediated effects were increased from 25% to 80% (more than 3-fold) suggesting its potential for pancreatic cancer treatment. These findings suggest that PDT in combination with DCA synergizes with the programmed cell death process. Figure 7B shows consolidated data based on Figure 7A.

We further investigated whether PDT-DCA-induced apoptosis led to activation of caspases linked to a mitochondria-mediated intrinsic signaling pathway. Flow cytometry was used to quantify caspase-3/-7 activation, a marker for intrinsic apoptosis. Figure 8A shows that the combined treatment of PDT with DCA significantly induced caspase-3/-7 activation in PANC-1 cells. While 30 mM DCA alone produced only 11% apoptotic cells, PDT with DCA at 54 J/cm^2^ nearly tripled this effect (29%). Increasing the laser dose to 108 J/cm^2^ further enhanced caspase-3/-7 activation to 37%. The results demonstrate that PDT in combination with DCA synergistically induced intrinsic apoptotic cell death in pancreatic cancer cells. Figure 8B shows consolidated data based on Figure 8A.

### 2.7. PDT-DCA Triggers Ferroptosis-Mediated Cell Death

Ferroptosis constitutes an iron-dependent regulated cell death modality defined by lipid peroxide accumulation, arising from iron-catalyzed peroxidation of polyunsaturated fatty acids (PUFAs) in cellular membranes, which culminates in membrane rupture [22]. Mitochondria are integral to ferroptosis execution, amplifying oxidative stress through reactive oxygen species (ROS) and lipid radical propagation. Given that PDT and DCA independently evoke ferroptosis and given our prior evidence of mitochondrial involvement in their synergy, we hypothesized that PDT-DCA would elicit mitochondria-driven ferroptosis in PANC-1 cells. Ferroptosis was quantified by flow cytometry detection of lipid peroxidation using BODIPY 581/591 C11, distinguishing non-oxidized (red fluorescence) from peroxidized (green fluorescence) lipids. Flow cytometry analysis revealed negligible ferroptosis induction with single agents (DCA or 5-ALA alone) relative to untreated controls (Figure 9A). Dual 5-ALA-DCA and laser–DCA combinations modestly elevated ferroptosis to 12.4% and 28.6%, respectively. Strikingly, PDT-DCA synergistically induced ferroptosis in 72.8% of cells at 54 J/cm^2^ laser fluence, escalating to 96.5% at 108 J/cm^2^.

### 2.8. Bliss Independence Analysis Reveals That DCA Synergizes with PDT Effects

The combination of ALA (at 54 J/cm^2^) with DCA resulted in moderate-to-strong synergistic interactions across multiple cell death pathways (Table 1). The analysis indicated that this combination most potently enhanced oxidative stress, ferroptotic signaling, and ATP release, while the activation of executioner caspases remained largely additive. Increasing the laser fluence to 108 J/cm^2^ in the presence of the ALA + DCA combination further enhanced the synergistic profile. As shown in Table 2, The ALA + 108 + DCA doses produced stronger Bliss Scores across apoptosis, ferroptosis, ROS generation, and ATP release compared with the ALA + 54 + DCA combination, indicating that higher laser fluence intensifies metabolic and oxidative stress in cancer cells. In summary, while both laser fluences produced synergistic effects in combination with ALA and DCA, the ALA + 108 + DCA combination consistently yielded higher Bliss Scores. This suggests a laser-dose-dependent enhancement in synergistic cell death. The most pronounced cooperative effects were observed in ROS elevation, ferroptotic signaling, and ATP depletion, suggesting that the underlying mechanisms of synergy are centered on metabolic collapse and oxidative damage. In contrast, caspase-3/-7 activation remained near-additive under both conditions.

## 3. Discussion

Pancreatic ductal adenocarcinoma (PDAC), exemplified by the PANC-1 cell line in this study, is an aggressive cancer with resistance to conventional chemotherapies. No single-agent therapy has proven effective [1]. Combination therapies targeting multiple oncogenic pathways offer a promising approach to overcoming PDAC’s chemoresistance and immunotherapy. Photodynamic therapy (PDT), using 5-aminolevulinic acid (5-ALA), induces apoptosis, damages tumor vasculature, and activates anti-tumor immunity through reactive oxygen species (ROS) generated upon light activation [23,24]. Combining PDT with dichloroacetic acid (DCA), a metabolic modulator which is known to shift anaerobic glycolysis to mitochondrial oxidative phosphorylation, induces apoptosis and inhibits cell proliferation [21]. To our knowledge, this unique combination therapy has not yet been studied in this context in pancreatic cancer, although individual effects of PDT and DCA are documented.

We hypothesized that combining PDT with DCA would target pancreatic cancer cells through mitochondrial dysfunction and metabolic rewiring. In PANC-1 cells, our findings suggest for the first time that PDT + DCA enhances three regulated cell death (RCD) pathways: apoptosis, immunogenic cell death (ICD), and ferroptosis. Bliss independence model analysis determined synergistic effects in most functional assays for the combined treatments (PDT + DCA) at two different PDT doses. Collectively, our results position mitochondria as a central hub where oxidative and metabolic stresses converge. PDT-induced ROS production disrupts mitochondrial membranes, causing mitochondrial membrane potential depolarization (Δψm) and caspase-3-/7 activation, consistent with prior reports [13,18,25]. DCA enhances this effect by inhibiting pyruvate dehydrogenase kinase (PDK), reversing the “Warburg” effect and promoting oxidative phosphorylation, which increases mitochondrial ROS and sensitizes cells to apoptosis [26,27]. Additionally, PDT + DCA induces ATP release, a hallmark of ICD, suggesting potential DAMP-mediated immunogenic signaling, pending further validation [28]. Lipid peroxidation assays confirmed involvement of ferroptosis, an iron-dependent RCD pathway driven by DCA’s metabolic shift, which reduces NADPH availability for glutathione peroxidase 4 (GPX4) antioxidant defense [9].

To select a safe dose of DCA to be used in PDT combination therapy, we first determined the IC_50_ value in concentration-dependent cytotoxicity assays. The IC_50_ of DCA aligns with studies in breast cancer [21] and glioblastoma supporting its broad spectrum of anticancer potential. The IC_50_ values for DCA in pancreatic cancer cell lines can range substantially dependent on specific cancer types [29]. PDT generates singlet oxygen and other ROS, causing acute mitochondrial outer-membrane permeabilization (Δψm) and endoplasmic reticulum (ER) stress, which trigger apoptosis and DAMP release for ICD [28,30]. DCA’s inhibition of PDK redirects pyruvate into oxidative phosphorylation, amplifying mitochondrial ROS and creating a redox imbalance that depletes antioxidant defenses, priming cells for ferroptosis [26,27]. ER stress from PDT further enhances ICD by promoting ATP release and calreticulin exposure, while DCA’s metabolic rewiring exacerbates mitochondrial collapse, linking all three RCD modalities. This convergence on mitochondria as a “final common pathway” underscores their role as gatekeepers of cell death in PDAC [18,30].

This study is limited to in vitro assays using the PANC-1 cell line, lacking immunocompetent models to validate ICD or pharmacological inhibitors (e.g., ferrostatin-1 and Z-VAD-FMK) to confirm specific RCD pathways. Given the high degree of heterogeneity in the PANC-1 cell line, the findings of this study require validation in other cell lines (e.g., Mia PaCa-2 and AsPC-1) in future research. Subsequent studies can then be expanded to include pancreatic cancer-derived organoids and in vivo models. In vivo validation of ATP release along with other DAMPs could confirm and validate the potential of immunogenic cell signaling for anti-tumor immunity. Glutathione and GPX4 activities were not quantified to fully elucidate DCA’s role in ferroptosis. Future studies should validate these mechanisms in vivo, optimize treatment parameters (e.g., photosensitizer delivery and light dosimetry), and explore synergy with immunotherapies like checkpoint inhibitors.

Pragmatically, an ideal anticancer therapy would be easily administered and affordable. Unfortunately, most of the current therapies, including novel cellular therapies, are prohibitively expensive for millions of patients around the globe. Thus, novel therapies that are less expensive are desirable. Beyond mechanistic advantages, PDT + DCA offers practical benefits, such as affordability. PDT is a minimally invasive therapeutic modality that is used in the management of various cancers, including lung, early gastric, and skin cancers [31]. Therefore, PDT + DCA offers a multifaceted approach to overcoming PDAC resistance by inducing apoptosis, ICD, and ferroptosis through mitochondrial dysfunction and oxidative stress. This combination’s ability to target multiple tumor survival programs highlights its potential as a novel therapeutic strategy, warranting further mechanistic and translational studies.

## 4. Materials and Methods

### 4.1. Materials

The PANC-1 cell line, established from a pancreatic carcinoma of ductal origin (PDAC; catalog # CRL-1469), as well as DMEM (catalog # 30-2002) and FBS (catalog # 30-2020), was purchased from American Type Culture Collection (ATCC, Manassas, VA, USA). CyQUANT™ LDH Cytotoxicity Assay-Kit (catalog # C20302) and BODIPY™ 581/591 C11 lipid peroxidation sensor (catalog # D3861) were obtained from Invitrogen (Carlsbad, CA, USA). The “Highly Stable ATP Assay Kit” (catalog # ab287863) was purchased from Abcam (Waltham, MA, USA). Flow Cytometry apoptosis assay kit (catalog # V13241) was purchased from Fisher Scientific (Hanover Park, IL, USA). Technical-grade chemicals including 5-ALA (catalog # A3785), DCA (catalog # D54702), JC-1 dye (catalog # T4069), DCFH-DA (catalog # D6883), 0.4% Trypan blue (catalog # T8154), MTT (catalog # 50-488-833), Penicillin–Streptomycin (catalog # 15140148), 0.25% trypsin–EDTA (catalog # 25200056), and PBS (catalog # P4474) were purchased from Sigma-Aldrich (St. Louis, MO, USA). The 635 nm laser was purchased from Opto Engine, LLC (Midvale, UT, USA).

### 4.2. Cell Culture

PANC-1 cells were maintained in complete DMEM containing 10% FBS with antibiotics (penicillin and streptomycin) at 37 °C in a 5% CO_2_ atmosphere. Cells were grown in logarithmic growth, and when the cell layer reached 80% confluence, they were trypsinized and subcultured as described before [32]. Briefly, confluent cells were washed once with PBS (pH 7.4) followed by treatment with trypsin–EDTA solution (0.05% trypsin/0.53 mM EDTA). After trypsinization, cells were collected and washed by centrifugation in PBS and subsequently resuspended in complete DMEM. These cells were then used for experiments. Cell density was adjusted to desired concentration after counting with a hemocytometer. Cell viability was checked by Trypan blue, and cells were added to micro-well plates for various experiments, which are detailed under each section.

### 4.3. MTT Cytotoxicity Assay

Cytotoxicity by dichloroacetic acid (DCA), if any, was assessed by the MTT assay as described previously [21]. Briefly, PANC-1 cells were seeded into 96-well plates at a density of 2.0 × 10^3^ cells/well in 100 µL of complete DMEM and incubated for 24 h at 37 °C in a 5% CO_2_ incubator to form a monolayer. The following day, cells were washed once, with medium being replaced with fresh medium, and incubated with varying concentrations of DCA; incubation continued for an additional 24 h. Vehicle controls were included for all assays. Following DCA treatment for 24 h, 10 µL of MTT reagent (5 mg/mL PBS) was added directly to each well without washing. Plates were gently agitated for 3 min to mix the MTT reagent and then incubated at 37 °C under dark conditions. After incubation for 4 h, 90 µL of supernatants was carefully aspirated off without disturbing the cell monolayer. Formazan crystals formed in the cells were solubilized by adding 100 µL of DMSO per well. Plates were shaken for 15 min at room temperature to ensure complete solubilization of formazan crystals. Absorbance was measured at 570 nm in a BioTek SYNERGY H4 hybrid microplate reader, Agilent Technologies, Inc., Santa Clara, CA, USA and results were expressed as percentages of the untreated control as described before [33]. Dose–response curves and IC_50_ values were generated using GraphPad Prism software (v 8.02) and the Quest Graph™ IC_50_ Calculator, AAT Bioquest, Inc. Pleasanton, CA, USA, respectively.

### 4.4. Lactate Dehydrogenase Assay

Lactate dehydrogenase (LDH) release from cells is a widely accepted marker for cellular damage [34]. For the assessment of cytotoxicity, the LDH release assay was performed using a CyQUANT™ LDH fluorescence cytotoxicity assay kit, ABCam, Cambridge, UK. For this, PANC-1 cells were seeded into 96-well plates at a density of 2.0 × 10^3^ cells/well in 100 µL of complete medium and incubated for at 37 °C to form a monolayer. After 24 h, cells were treated with different concentrations of DCA or vehicle control and incubated for an additional 24 h at 37 °C. At the end of the incubation period, 50 µL aliquots were transferred to black clear-bottom 96-well plates containing 50 µL of the assay kit reaction mixture. Contents in the wells were mixed gently by tapping and incubated for 10 min at room temperature. At this point, 50 μL of stop solution was added to each well, and the fluorescence emitted was read at 560 nm and 590 nm using a fluorescence microplate reader (BioTek SYNERGY H4, Hybrid Technology). LDH activity was calculated by subtracting the background as described earlier [32].

### 4.5. Cell Proliferation Assay

Besides the cytotoxicity assay, the MTT assay is also used widely for assessing cell proliferation. It functions by measuring cellular metabolic activity by quantifiable colored formazan product [35,36]. To assess changes in cell proliferation due to DCA treatment, cells were prepared and seeded into 96-well plates similar to the cytotoxicity assay. Briefly, PANC-1 cells were seeded at a density of 2.0 × 10^3^ cells/well in 100 µL of complete medium and incubated for 24 h to allow cells to form a monolayer. Cells were then treated with different concentrations of DCA or vehicle control and incubated for 24, 48, 72, and 96 h. At the end of each pre-determined incubation time, cell proliferation was measured in a microplate reader at 570 nm, and the results were expressed as a function of absorbance, as described before [32].

### 4.6. Photodynamic Therapy with Infrared Laser Irradiation

Photodynamic therapy (PDT) in our cell culture experiments was carried out using a modified protocol described previously by Alkarakooly et al. [21]. Effects of treatment with photosensitizer (5-ALA) alone, infrared laser alone, and the combined effects of 5-ALA and laser and of laser and DCA were studied. Briefly, PANC-1 cells were seeded in alternate wells in 96-well plates to avoid overlap of infrared laser treatment spillover. Each experimental well received a density of 2.0 × 10^3^ cells/well in 100 µL of complete DMEM and was allowed to form a monolayer as described before. The following day, the culture medium was aspirated off, and wells were gently washed twice with PBS. At this point, experimental wells received either single-drug treatment with 5-ALA alone and DCA alone or combination treatment (5-ALA + DCA), as described in the figure legends. Control cells received only vehicle control without any drug treatment.

To test the effect of laser treatment alone, cells were exposed to varying fluences of 9, 27, 54, and 108 J/cm^2^ using an infrared laser (Opto Engine LLC, Midvale, UT, USA) which gives a wavelength of 635 nm. This laser source has a power of 16 mW. The calculated surface area for the spot size for irradiation was 0.3217 cm^2^, which gives an irradiance of 49.74 mW/cm^2^. This irradiance was maintained through all irradiation time periods to achieve fluences of 9, 27, 54, and 108 J/cm^2^. Untreated control cells were not irradiated and received no drug treatment.

For PDT experiments, cells were incubated with freshly prepared 5-ALA at concentrations of 0.5, 1.0, 1.5, or 2.0 mM in serum-free medium for 4 h at 37 °C. At the end of 4 h incubation with 5-ALA, cells were exposed to infrared laser fluences of 9, 27, 54, or 108 J/cm^2^. This constituted the standard PDT protocol (5-ALA + laser) that we followed in our experimental model. After irradiation, cells were washed with PBS and incubated with or without DCA for 24 h. After incubation, cell viability was assessed using the MTT assay as described before. For additional experiments using combined treatments, 5-ALA was used at a pre-determined concentration of 1.0 mM and laser dosing of 54 and 108 J/cm^2^. In experiments where the combined effect of PDT + DCA was assessed, PDT treatment was carried out as described before, followed by 30 mM DCA in 100 µl of complete DMEM. At the end of each experiment, cell viability/cytotoxicity was assayed by MTT as described before, and these experiments were repeated three times [21].

### 4.7. Determination of Mitochondrial Membrane Potential

Pancreatic Cancer Stem Cells (PCSCs) strongly rely on mitochondrial metabolism to maintain their stemness, thus representing a putative target for their elimination [15,37]. In this regard, mitochondrial membrane potential (MMP; ΔΨm) is a critical indicator of integrity of mitochondria, oxidative status, and cellular health. Indirectly, measurement of MMP reflects ROS production, ATP synthesis, apoptosis induction, and possibly other forms of cell death [38].

In this study, we focused on cellular metabolism and mitochondrial integrity to better understand the efficacy of PDT + DCA combinatorial treatment. To assess MMP, PANC-1 cells were seeded at 1 × 10^4^ cells/well in 96-well plates and treated as described in the ROS assay protocol. Post-treatment, cells were washed twice with PBS and incubated in serum-free medium containing 10 µg/mL JC-1 dye (5,5′,6,6′-tetrachloro-1,1′,3,3′-tetraethyl-benzimidazolyl-carbocyanine iodide) for 30 min at 37 °C in a 5% CO_2_ atmosphere. In healthy cells, the electrochemical gradient drives JC-1 accumulation in the mitochondrial matrix, forming red fluorescent J-aggregates (excitation: 560 nm; emission: 595 nm). Depolarized mitochondria exhibit reduced J-aggregate formation, increasing green-fluorescent monomers (excitation: 485 nm; emission: 535 nm). The red-to-green fluorescence ratio, calculated from fluorescence intensities, indicates MMP status, with a decreased ratio signifying depolarization. In this set of experiments, MMP was monitored by assessing fluorescence emitted from the cells using a BioTek Synergy H4 microplate reader. Data were analyzed in triplicate, normalized to untreated controls, and expressed as means ± SDs.

### 4.8. Determination of Reactive Oxygen Species

The 2′,7′-dichlorodihydrofluorescein diacetate (DCFH-DA) assay is a common method for measuring intracellular ROS in cells [39,40]. When incubated, the non-polar DCFH-DA freely diffuses across the cell membrane and enters the cell. Upon exposure to intracellular ROS, the non-fluorescent DCFH-DA is activated, forming the highly fluorescent compound 2′,7′-dichlorofluorescein (DCF). The fluorescence intensity is directly proportional to the amount of ROS present in the cell [39]. In this study, PANC-1 cells were plated in black 96-well plates with clear bottom at a density of 2.0 × 10^3^ cells/well in 100 µL of complete medium. As a standard procedure involving PDT experiments, only alternate wells received cells and grown for 24 h, as described before. Cells were then subjected to PDT with or without 5-ALA. Wells received single-drug treatment or combined treatment with DCA as outlined in the figure legends. At the end of the DCA treatment, DCFH-DA was added to the medium at a final concentration of 10 μM and incubated for 90 min in the dark. Subsequently, cells were washed twice with PBS and resuspended in 100 μL of PBS. The fluorescence intensity of DCF was measured using a fluorescence microplate reader (BioTek SYNERGY H4, Hybrid Technology) at excitation and emission wavelengths of 485 nm and 535 nm, respectively.

### 4.9. ATP Release Assay

In the context of cancer therapy, ATP release is a critical marker of immunogenic cell death (ICD), a process that activates adaptive immunity to target and eliminate malignant cells [41]. Living cells maintain a high intracellular concentration of ATP (1–10 mM) compared with the low, nanomolar levels normally found outside cells [42]. Cellular damage or death, particularly through regulated cell death pathways, causes shift in the concentration gradient, resulting in the release of a massive amount of ATP outside the cell, where it acts as a DAMP [43]. The ATP release assay was performed in PANC-1 cells after exposure to single-agent or combinational treatment as previously reported [32]. Briefly, a monolayer of PANC-1 cells in a 96-well plate was treated as described in the PDT-DCA combination experiments. At the end of the incubation periods, cell culture medium was collected to determine the ATP released. ATP concentrations were determined using the “Highly Stable ATP Assay Kit” purchased from Abcam following the manufacturer’s recommendations. Briefly, 90 μL of the reaction mix from the kit was aliquoted into a series of wells in a new 96-well plate for the number of standards and samples from the treatments. Ten microliters (10 μL) of standard samples or treated samples was added into the respective wells and mixed, and luminescence was detected using a microplate reader (SYNERGY H4, BioTek, Hybrid Technology). ATP concentrations in treated samples were calculated using the standard curve following the manufacturer’s instructions.

### 4.10. Apoptosis Detection by Flow Cytometry

Apoptosis was quantified using flow cytometry with Annexin V-FITC and propidium iodide (PI) staining. PANC-1 cells were seeded at 6.0 × 10^5^ cells/well in 6-well plates, cultured in DMEM with 5% CO_2_ at 37 °C for 24 h. Cells were then treated with DCA alone or combined with PDT as described above. Cells were harvested, washed with PBS, and resuspended in 100 µL of 1 × Annexin-Binding Buffer at 5 × 10^5^ cells. Samples were stained with 5 µL of Annexin V-FITC (green fluorescence, detecting early apoptosis) and 1 µL of PI (100 µg/mL, red fluorescence, marking late apoptosis/necrosis), incubated for 15 min in the dark, and diluted with 400 µL of buffer. Fluorescence was analyzed within 1 h using a BD LSRFortessa™ flow cytometer, BD Biosciences, San Jose, CA, USA, with emissions measured at 515–545 nm (FITC, FL-1) and 564–606 nm (PI, FL-2). Data were expressed as percentages of live (low fluorescence), early apoptotic (high green fluorescence), late apoptotic (green and red fluorescence), and necrotic (high red fluorescence) cell populations.

### 4.11. Caspase-3/-7 Activity

Caspase-3/-7 activity was used as an apoptotic marker. This is an indication of activation of an intrinsic apoptosis pathway involving mitochondria. PANC-1 cells were seeded into a 6-well microplate at a density of 6.0 × 10^5^ cells/well in 500 μL of complete DMEM and grown for 24 h at 37 °C in a 5% CO_2_ incubator to form a monolayer. Following incubation, the culture medium was aspirated, and the cells were gently washed twice with PBS. Afterwards, the cells were treated as described in the flow cytometry experiment. Cells were then harvested and resuspended in 500 µL of PBS. One microliter (1 µL) of CellEvent^TM^ Caspase-3/7 green fluorescence detection reagent was added to each 500 µL of cells and incubated for 60 min at room temperature. One microliter (1 µL) of 1 mM SYTOX™ AADvanced™ Dead Cell Stain was gently added during the last 5 min of staining. The samples were analyzed using the BD LSRFortessa™ flow cytometer at the UAMS Flow Cytometry Core Facility at 488 nm excitation and 530/30 and 690/50 band pass filters. Data in the flow cytometry histograms are presented as percentages of apoptotic cells.

### 4.12. Detection of Ferroptosis

PANC-1 cells were seeded in a 6-well microplate at a density of 6.0 × 10^5^ cells/well in 500 μL of complete DMEM and grown at 37 °C in a 5% CO_2_ incubator for 24 h to form a monolayer. Following incubation, the culture medium was aspirated, and the cells were gently washed twice with PBS. Afterwards, the cells were treated with DCA alone or with PDT and DCA as reported in the flow cytometry experiments. Cells were then washed twice with PBS. To evaluate lipid peroxidation, PANC-1 cells were stained with 2 µM C11-BODIPY 581/591 in serum-free DMEM for 30 min at 37 °C in the dark. After PBS washing and harvesting with trypsin–EDTA, cells were resuspended in flow cytometry buffer. Flow cytometry at 488 nm excitation was used to measure fluorescence emission in the FITC channel (oxidized dye, 510 nm, green) and phycoerythrin (PE) channel (non-oxidized dye, 590 nm, red). Ferroptosis was quantified as the percentage of cells showing change from red to green fluorescence, indicating lipid peroxidation.

### 4.13. Bliss Independence Interaction Analysis

Drug interactions were assessed using the Bliss independence model. For each assay, including apoptosis, caspase-3/-7 activity, mitochondrial membrane potential (MMP), reactive oxygen species (ROS) production, ATP release, and ferroptosis, the treatment responses were first normalized to the untreated control and expressed as fractional effects ranging from 0 to 1. The fractional effect of DCA monotherapy was defined as *E*_DCA_. The effects of ALA at the laser fluences of 54 J/cm^2^ and 108 J/cm^2^ were defined as *E*_PDT1_ and *E*_PDT2_, respectively. The experimentally measured effect of the combination treatment (ALA + Laser + DCA) was denoted by *E*_Observed_. The expected effect (*E*_Expected_) for an additive interaction was calculated according to the Bliss independence principle using the equation $E_{\mathrm{Expected}}=E_{DCA}+E_{PDT}-\left(E_{DCA}\times E_{PDT}\right).$ The Bliss Synergy Score was subsequently derived as $\mathrm{Bliss}\;\mathrm{Score}=E_{\mathrm{Observed}}-E_{\mathrm{Expected}}$. A positive Bliss Score indicates a synergistic effect, a score close to zero indicates an additive effect, and a negative score signifies antagonism. All calculations were performed using a customized Bliss Prism template in GraphPad Prism, which automatically computed *E*_Expected_ and Bliss Scores from the normalized fractional effect data.

### 4.14. Statistical Analysis

All cell-based assays in multi-well plates were performed in triplicate, and experiments were repeated independently at least three times. All data that show error bars are presented as means ± SE unless otherwise mentioned. GraphPad Prism Version 8.02 (GraphPad Software Inc., San Diego, CA, USA) was used to analyze the data. To calculate statistically significant differences between the test and control, a two-way ANOVA with post hoc Tukey’s test was employed. Statistical significance was defined as * *p* ≤ 0.05, ** *p* ≤ 0.01, and *** *p* ≤ 0.001. IC_50_ values were determined using the Quest Graph™ IC_50_ Calculator (Quest Graph, v 6.0, AAT Bioquest, Inc., Pleasanton, CA, USA). For flow cytometry experiments, data presented are representative histograms of three independent experiments performed in triplicate. Data in bar graphs represent means ± SE from three independent experiments performed in triplicate. A Student’s *t*-test was used to determine statistically significant differences between control and experimental values (* *p* ≤ 0.05, ** *p* ≤ 0.01, and *** *p* ≤ 0.001).

## Figures and Tables

**Figure 1 ijms-26-11031-f001:**
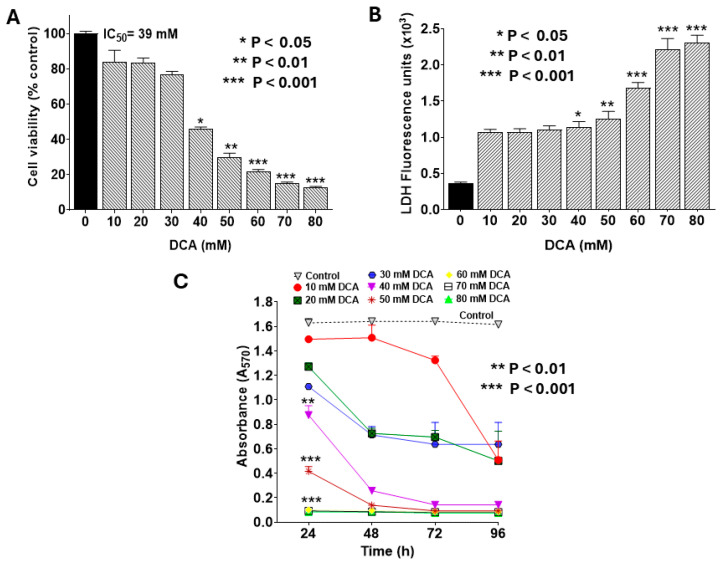
Dose-dependent effects of DCA on PANC-1 cell viability and proliferation. PANC-1 cells were seeded in 96-well plates at 2.0 × 10^3^ cells/well in complete DMEM and incubated for 24 h at 37 °C to form monolayers. Cells were then treated with DCA at varying concentrations for the indicated times. Assays for cell viability, LDH release, and proliferation were performed as detailed in the Materials and Methods Section. (**A**) Cell viability was assessed by MTT assay following 24 h DCA treatment. Data were analyzed using GraphPad Prism v8.02, with IC_50_ determined using Quest Graph™ IC_50_ Calculator (AAT Bioquest). (**B**) LDH release into culture medium was quantified using CyQUANT™ LDH Cytotoxicity Assay Kit. LDH activity was calculated by subtracting background fluorescence from untreated controls and expressed in U/L. (**C**) Cell proliferation was monitored at 24, 48, 72, and 96 h post-DCA treatment using MTT assay. Data represent mean values ± SE (standard error) from at least 3 independent experiments, each performed in quadruplicate. The statistically significant difference and *p*-values were determined for all time points. However, the asterisks are shown only for the 24 h treatment for the sake of clarity.

**Figure 2 ijms-26-11031-f002:**
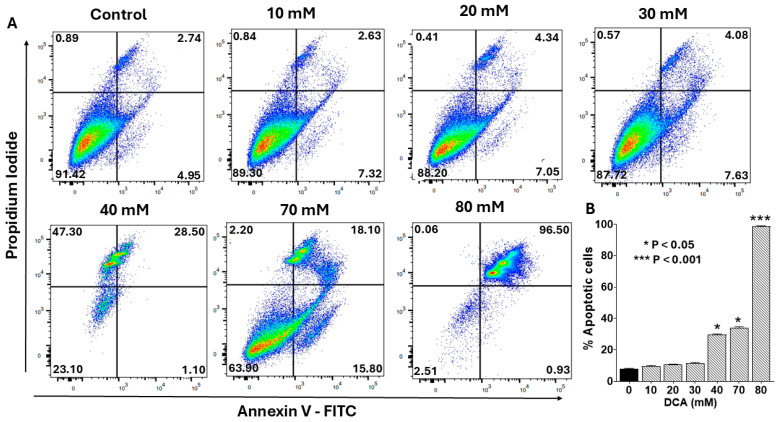
Flow cytometry analysis of dose-dependent effect of DCA on apoptosis. PANC-1 cells were seeded in 6-well plates at a density of 6.0 × 10^5^ cells/well in complete DMEM and grown for 24 h at 37 °C in a 5% CO_2_ incubator to form monolayers. Cells were then treated with the indicated concentrations of DCA for an additional 24 h. Control cells were treated with complete media only. (**A**) Cells were harvested and stained with Annexin V Alexa Fluor™ 488 and propidium iodide and analyzed by flow cytometry as described in the Materials and Methods Section. Histograms shown are representative of three independent experiments performed in triplicate. (**B**) Data in bar graph represent means ± SE from three independent experiments, each performed in triplicate. A Student’s *t*-test was used to determine statistically significant differences between control and experimental values. Asterisks denote significant differences compared with the control.

**Figure 3 ijms-26-11031-f003:**
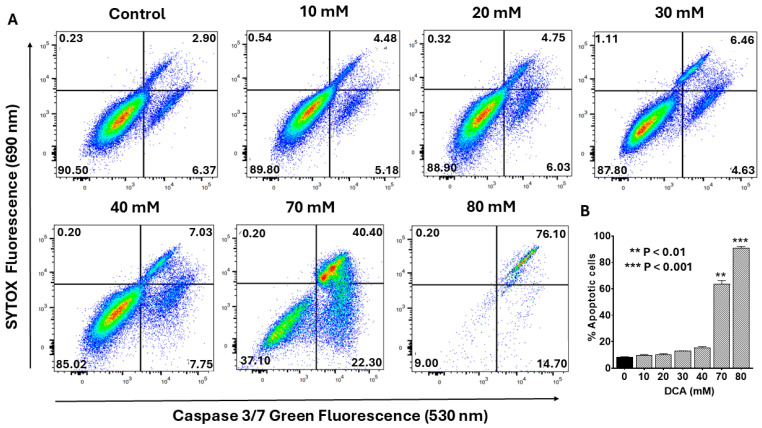
Flow cytometry analysis of dose-dependent effect of DCA on caspase-3/-7 activation. PANC-1 cells were seeded in 6-well plates at a density of 6.0 × 10^5^ cells/well in complete DMEM and grown for 24 h at 37 °C to form monolayers. Cells were then treated with the indicated concentrations of DCA for an additional 24 h. Control cells were treated with media only. (**A**) Activation of caspase-3/-7 was measured in control and experimental wells using a flow cytometry kit containing CellEvent™ Caspase-3/7 Green Detection Reagent and SYTOX™ Advanced Dead Cell Stain. Cells were analyzed by flow cytometry as described in the Materials and Methods Section. Histograms shown are representative of three independent experiments performed in triplicate. (**B**) Data in bar graph represent means ± SE from three independent experiments, each performed in triplicate. A Student’s *t*-test was used to determine statistically significant differences between control and experimental values. Asterisks denote significant differences compared with the control.

**Figure 4 ijms-26-11031-f004:**
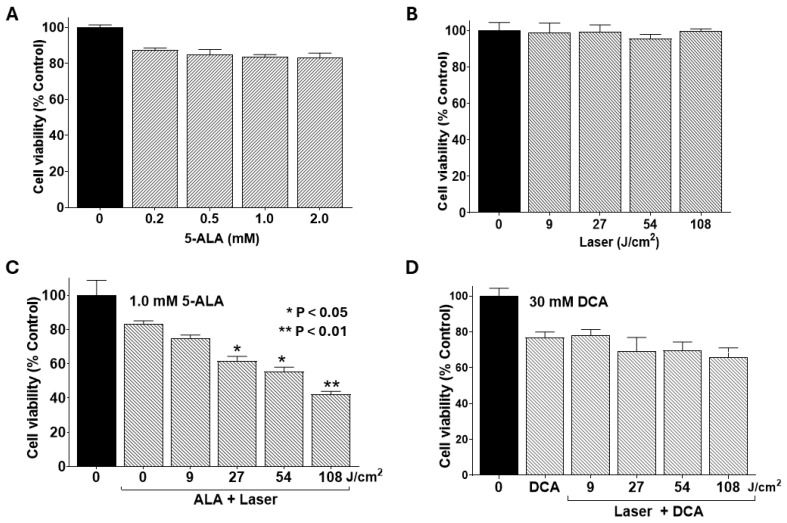
Effects of 5-ALA and laser alone and combinations with DCA on cell viability. PANC-1 cells were seeded into 96-well plates at a density of 2.0 × 10^3^ cells/well in complete DMEM and grown for 24 h at 37 °C to form monolayers. Cells were then treated with a single agent or in combination with DCA, and cell viability was determined using MTT assay as described in the Materials and Methods Section. (**A**) Effect of 5-ALA treatment alone at the indicated concentrations for 4 h. (**B**) Effect of laser irradiation alone at the indicated fluences. (**C**) Single-agent treatment, as well as the effect of PDT, a combination of 1.0 mM 5-ALA for 4 h, and the indicated fluences of laser irradiation. (**D**) Combined effect of laser irradiation at the indicated fluences (without 5-ALA treatment) followed by treatment with sub-toxic dose of 30 mM DCA for 24 h. Data shown are means ± SE from three independent experiments, each performed in quadruplicate. Asterisks denote significant differences compared with the control.

**Figure 5 ijms-26-11031-f005:**
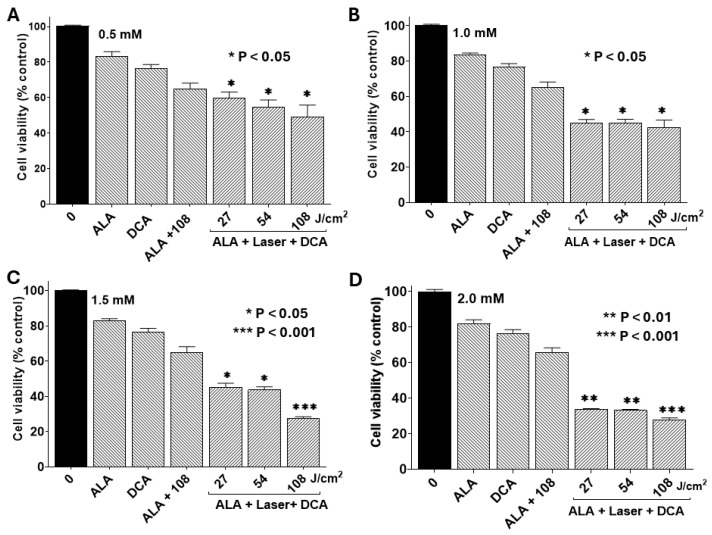
Combined effects of PDT and DCA on cell viability. PANC-1 cells were seeded into 96-well plates at a density of 2.0 × 10^3^ cells/well in complete DMEM and grown for 24 h at 37 °C to form monolayers. Cells were then incubated at 37 °C with different concentrations of 5-ALA for 4 h. Dose-dependent effects of 5-ALA were assessed in four separate sets of experiments: (**A**) 0.5 mM 5-ALA, (**B**) 1.0 mM 5-ALA, (**C**) 1.5 mM 5-ALA, and (**D**) 2.0 mM 5-ALA. After 5-ALA treatment, cells were washed and irradiated with a 635 nm laser at fluence of 27, 54, or 108 J/cm^2^ as indicated. After laser exposure, cells were incubated with or without 30 mM DCA for an additional 24 h to examine the effect of combination therapy. Appropriate controls were run simultaneously. Cell viability was determined by MTT assay, and data were calculated as percentages of live cells in comparison with the untreated control. Data are presented as means ± SE from three independent experiments, each performed in quadruplicate. Asterisks denote values significantly different from the controls.

**Figure 6 ijms-26-11031-f006:**
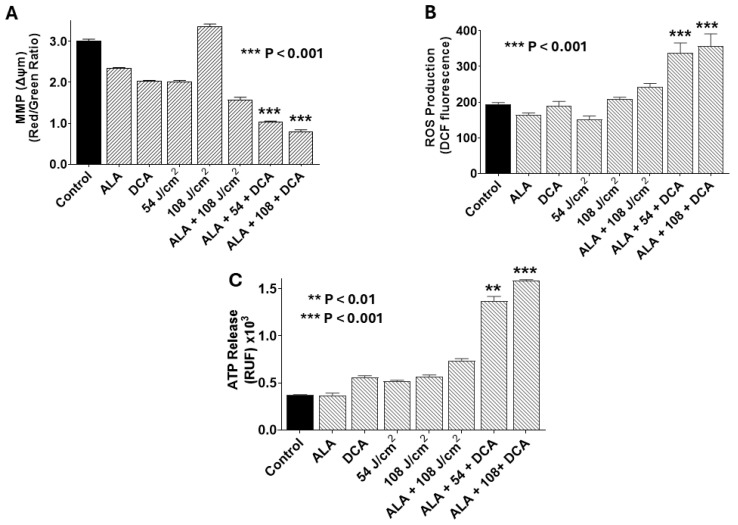
PDT-DCA disrupts mitochondrial membrane potential (MMP), elevates reactive oxygen species (ROS) production, and induces ATP release in PANC-1 cells. Cells were seeded in 96-well plates at 2 × 10^3^ cells/well and cultured for 24 h at 37 °C to form monolayers. Treatments included (i) 5-ALA only (1.0 mM) for 4 h, PBS wash, and incubation continued for an additional 24 h in fresh DMEM; (ii) DCA only (30 mM for 24 h); (iii) 635 nm infrared laser only at 54 or 108 J/cm^2^); (iv) 5-ALA + DCA (1.0 mM 5-ALA for 4 h, PBS wash, and then 30 mM DCA for 24 h without irradiation); and (v) PDT-DCA (1.0 mM 5-ALA for 4 h, followed by laser irradiation at 54 or 108 J/cm^2^, and then 30 mM DCA for 24 h). Untreated cells served as controls. Post-treatment, cells were assessed as follows: (**A**) MMP (Δψm) was determined using JC-1 staining (30 min at 37 °C), with depolarization being reflected by reduced red-to-green fluorescence ratio (measured via fluorescence plate reader). (**B**) Intracellular ROS production measured using 10 μM DCFH-DA staining (90 min in the dark), with fluorescence being quantified (excitation/emission: 485/535 nm) using a microplate reader post-wash. (**C**) Extracellular ATP was measured via bioluminescence assay (marker of ICD), interpolated from an ATP standard curve. Data represent means ± SE from three independent experiments, each performed in quadruplicate. Asterisks denote significant differences compared with the control.

**Figure 7 ijms-26-11031-f007:**
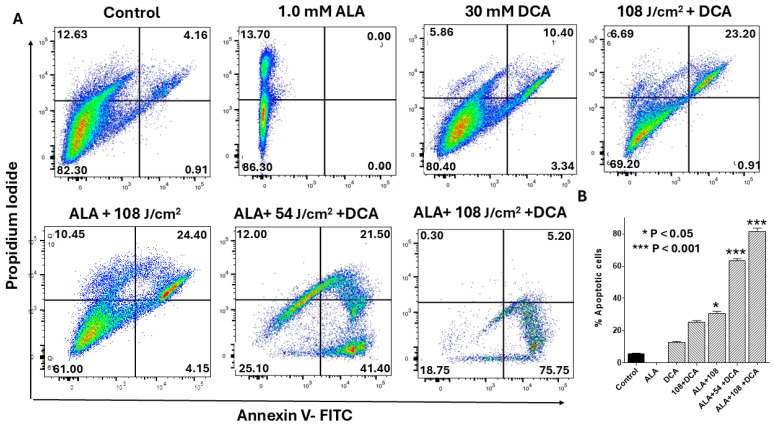
Flow cytometry analysis of synergistic effects of PDT-DCA on apoptosis. PANC-1 cells were seeded in 6-well plates at a density of 6.0 × 10^5^ cells/well in complete DMEM and grown for 24 h at 37 °C to form monolayers. Cells were then incubated with 1.0 mM 5-ALA for 4 h, irradiated (635 nm laser at 54 or 108 J/cm^2^), and treated with 30 mM DCA for 24 h as described in the Methods Section. Controls and single-agent treatment groups were included. (**A**) Representative scatter plots illustrate dose-dependent apoptotic shifts (early: Annexin V-positive/PI-negative, lower right quadrant; late: Annexin V-positive/PI-positive, upper right quadrant). With 108 J/cm^2^ PDT, DCA elevated total apoptosis from ~25% (PDT alone) to ~80% (>3-fold increase), highlighting marked synergy. (**B**) Data in bar graph represent means ± SE from three independent experiments, each performed in triplicate. A Student’s *t*-test was used to determine statistically significant differences between control and experimental values. Asterisks denote significant differences compared with the control.

**Figure 8 ijms-26-11031-f008:**
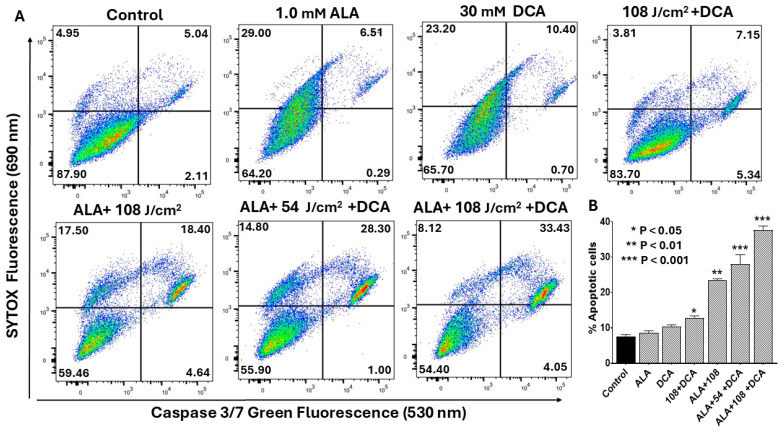
Flow cytometric analysis of PDT-DCA-induced caspase-3/-7 activation. PANC-1 cells were seeded in 6-well plates at a density of 6.0 × 10^5^ cells/well in complete DMEM and grown for 24 h at 37 °C to form monolayers. Cells were then subjected to (PDT + DCA) combination therapy as follows: Cells were treated with 1.0 mM 5-ALA for 4 h at 37 °C, followed by laser irradiation at fluences of 54 or 108 J/cm^2^. After irradiation, cells were washed with PBS and treated with 30 mM DCA for an additional 24 h. Control groups were treated with laser or DCA alone, or with ALA and DCA only. (**A**) Caspase-3/-7 activation was measured by a flow cytometry method using CellEvent™ Caspase-3/7 Green Detection Reagent and SYTOX™ Advanced Dead Cell Stain as described in the Materials and Methods Section. Histograms shown are representative of three independent experiments performed in triplicate. (**B**) Data in bar graph represent means ± SE from three independent experiments, each performed in triplicate. A Student’s *t*-test was used to determine statistically significant differences between control and experimental values. Asterisks denote significant differences compared with the control.

**Figure 9 ijms-26-11031-f009:**
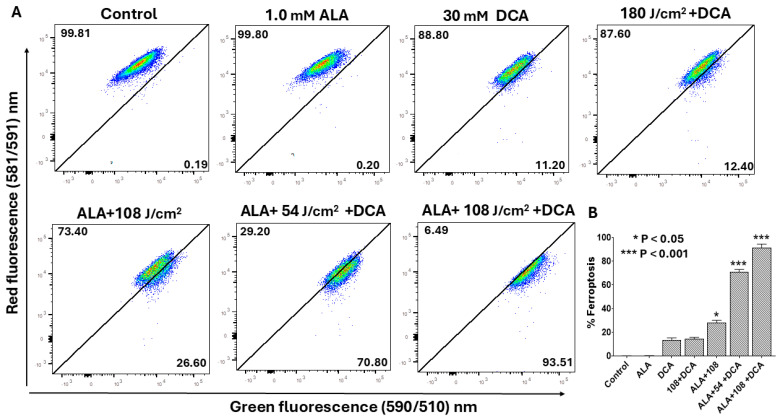
Flow cytometry assessment of ferroptosis induction by PDT-DCA in PANC-1 cells. Cells were seeded in 6-well plates at 6 × 10^5^ cells/well in complete DMEM and incubated for 24 h at 37 °C to establish monolayers. For PDT + DCA treatment, cells were first incubated with 1.0 mM 5-ALA for 4 h at 37 °C, followed by 635 nm laser irradiation (54 or 108 J/cm^2^). At the end of the incubation period, cells were washed in PBS and then treated with 30 mM DCA in fresh medium for 24 h. Controls included as single-agent treatment or PDT alone. Controls did not receive DCA in the culture wells. (**A**) Lipid peroxidation was quantified post-treatment via flow cytometry: cells were stained with 2 μM C11-BODIPY 581/591 for 30 min at 37 °C, harvested, and analyzed as described in the Materials and Methods Section, monitoring fluorescence shift from red (non-oxidized) to green (peroxidized). Shown are representative histograms depicting dose-dependent ferroptosis escalation from three independent experiments, each performed in replicate. (**B**) Data in bar graph represent means ± SE from three independent experiments, each performed in triplicate. A Student’s *t*-test was used to determine statistically significant differences between control and experimental values. Asterisks denote significant differences compared with the control.

**Table 1 ijms-26-11031-t001:** PDT1 + DCA: Bliss independence analysis showing synergistic effects at 54 J/cm^2^ laser exposure in the presence of 1.0 mM 5-ALA (PDT1) and 30 mM DCA in most functional assays.

	*E* _DCA_	*E* _PDT1_	Bliss Observed PDT1 + DCA	Bliss Expected PDT1 + DCA	Bliss Score PDT1 + DCA	Bliss Interpretation
Apoptosis	0.1260	0.2061	0.6335	0.3061	0.3274	Strong synergy
Caspases-3/-7	0.1036	0.1342	0.2796	0.2240	0.0557	Weak synergy
Ferroptosis	0.1353	0.1400	0.7061	0.2564	0.4498	Very strong synergy
ROS	0.0000	0.2140	0.8807	0.2140	0.6667	Strong synergy
ATP	0.1545	0.2012	0.8195	0.3246	0.4950	Strong synergy
MMP	0.3272	0.2709	0.6582	0.5095	0.1487	Mild synergy

**Table 2 ijms-26-11031-t002:** PDT2 + DCA: Bliss independence analysis showing stronger synergistic effects at 108 J/cm^2^ laser exposure in the presence of 1.0 mM 5-ALA (PDT2) and 30 mM DCA in most functional assays.

	*E* _DCA_	*E* _PDT2_	Bliss Observed PDT2 + DCA	Bliss Expected PDT2 + DCA	Bliss Score PDT2 + DCA	Bliss Interpretation
Apoptosis	0.1260	0.3061	0.8158	0.3935	0.4223	Stronger synergy
Caspases-3/-7	0.1036	0.2337	0.3771	0.3131	0.0639	Mild synergy
Ferroptosis	0.1353	0.2800	0.9133	0.3774	0.5359	Stronger synergy
ROS	0.0000	0.3012	1.0000	0.3012	0.6988	Peak synergy
ATP	0.1545	0.3010	1.0000	0.4090	0.5910	Very strong synergy
MMP	0.3272	0.4793	0.7347	0.6497	0.0850	Mild synergy

## Data Availability

The original contributions presented in this study are included in the article. Further inquiries can be directed to the corresponding author(s).

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
