# Peer review of "Dichloroacetic Acid Enhances Photodynamic Therapy-Induced Regulated Cell Death in PANC-1 Pancreatic Cancer Cell Line"

_ijms, 2025, doi:10.3390/ijms262211031_

Round 1

Reviewer 1 Report

Comments and Suggestions for Authors

This is a well-written manuscript that addresses a critical need for novel therapeutic strategies against pancreatic cancer. This manuscript investigates the combination of Dichloroacetic Acid (DCA) and Photodynamic Therapy (PDT) as a potential treatment for pancreatic cancer, using the PANC-1 cell line as an in vitro model. The authors report that DCA enhances the cytotoxic effects of PDT, and they explore the underlying mechanisms, suggesting the involvement of apoptosis, immunogenic cell death (ICD), and ferroptosis. The topic is of interest, as novel therapeutic strategies for pancreatic cancer are urgently needed. However, the manuscript has several major issues related to data presentation, statistical analysis, and the substantiation of its central claims that must be addressed before it can be considered for publication.

Major Concerns:

  1. The use of the PANC-1 cell line is an excellent and well-justified starting point for this investigation. To further strengthen the study's conclusions and demonstrate that this compelling effect is not cell-line specific, the authors might consider validating their key findings in a second pancreatic cancer cell line. Given the heterogeneity of PDAC, showing that the DCA-mediated enhancement of PDT holds true in another model (e.g., Mia PaCa-2 or AsPC-1) would significantly broaden the potential impact of this therapeutic strategy.
  2. A key finding of the manuscript is the enhanced efficacy of combining DCA and PDT. The authors compellingly describe this as a synergistic effect. To provide robust, quantitative support for this conclusion, it would be highly beneficial to perform a formal drug interaction analysis. Calculating a Combination Index (CI) using the Chou-Talalay method, or applying a Bliss Independence model, would definitively distinguish between a synergistic and a merely additive effect. This addition would provide a powerful quantitative foundation for the central claim of the paper.
  3. The investigation into the underlying cell death mechanisms is a highlight of the manuscript. The data suggesting the involvement of apoptosis, immunogenic cell death (ICD), and ferroptosis is intriguing. To make these conclusions even more definitive, the authors could consider a few additional experiments. For instance, using specific inhibitors (e.g., the pan-caspase inhibitor Z-VAD-FMK for apoptosis, or ferrostatin-1 for ferroptosis) could demonstrate that these pathways are functionally required for the observed cell death. Further, supplementing the current ICD and ferroptosis data with more direct assays (e.g., surface calreticulin staining for ICD or a lipid peroxidation assay for ferroptosis) would substantially bolster these already interesting findings.

    Without these functional experiments, it may be advisable to slightly moderate the language in the conclusion. Phrasing such as "our results suggest the involvement of..." rather than "we delineated the mechanism..." would more accurately reflect the current, very promising, dataset.

  4. A significant flaw throughout the manuscript is the absence of error bars and statistical tests for experiments described as replicates. The authors state that data in Figures 2b, 3b, 7b, 8b, and 9b are consolidated from triplicate repeats, yet no error bars (e.g., SD or SEM) are shown. Without these, the variability of the data and the reliability of the results cannot be assessed. Furthermore, no statistical tests are mentioned or applied to determine if the observed differences are significant. This is a critical omission that undermines the conclusions drawn from these experiments. All data derived from experimental repeats must be presented with error bars, and appropriate statistical tests must be performed and clearly described in the methods and figure legends.
  5. The manuscript's rationale could be clarified. DCA is known to have its own toxicity profile. The narrative would be more compelling if it first established a clear need for improvement—for instance, by demonstrating that PANC-1 cells are relatively resistant to PDT alone at clinically achievable parameters. The study could then present the combination with DCA as a targeted strategy to overcome this resistance. Recommendation: Consider re-framing the introduction and the presentation of results to first highlight the limitations of PDT in this context, thereby strengthening the argument for adding DCA. Is the goal to improve PDT, or is the combination meant to leverage the distinct mechanisms of both treatments? Clarifying this would improve the manuscript's impact.

Minor Concerns

  1.  The introduction is somewhat lengthy and could be made more concise. It should be streamlined to focus more directly on the problem of pancreatic cancer treatment resistance, the role of cellular metabolism (as targeted by DCA), the mechanism of PDT, and a clear statement of the study's hypothesis.
  2. Clarity of Significance Indicators in Figure 1a: The use of horizontal lines and asterisks to denote statistical significance in Figure 1a is confusing. The lines appear to indicate a comparison between adjacent bars (e.g., 50 mM vs. 60 mM DCA), whereas the comparison should be between each treatment and the control (0 mM DCA). A clearer method would be to place an asterisk directly above each bar that is significantly different from the control, with the significance level (p-value) defined in the figure legend.
  3. Figure 1c displays time-series data as bar charts for each time point. This format makes it difficult to visualize the trend of cell viability over time for each concentration. This data would be much more informative if plotted as a series of line graphs (time curves), with time on the x-axis, cell viability on the y-axis, and a separate line for each DCA concentration. This would allow for a much clearer comparison of the dynamic effects.
  4. The methods are clearly described. To further enhance reproducibility, it would be helpful to include a few more specific details about the PDT protocol. Explicitly stating the light dose (fluence, in J/cm²), the power of the light source (irradiance, in mW/cm²), and the specific wavelength of light used in the methods section. Additionally, it was noted that the light source was specified as 800 nm. Could the authors please confirm this wavelength? The active photosensitizer derived from 5-ALA, Protoporphyrin IX, is most efficiently activated by light in the 630-635 nm range. A brief rationale for selecting the 800 nm wavelength, if it was indeed used, would be a valuable clarification for the reader.

The manuscript addresses an important problem and presents an interesting therapeutic combination. However, in its current form, the lack of rigorous statistical analysis and data representation prevents a clear assessment of the results. The claims of synergy and the activation of multiple specific cell death pathways need stronger, more direct evidence. Addressing the major and minor points detailed above would significantly strengthen the manuscript.

Reviewer 2 Report

Comments and Suggestions for Authors

The manuscript reported the enhanced inducing effect of Dichloroacetic Acid on Photodynamic Therapy by the Regulated Cell Death in Pancreatic Cancer Cells, and detected the possible mechanisms of the bioactivities such as the mitochondria-mediated apoptosis, immunogenic cell death (ICD), and ferroptosis. But there have obvious shortcomings in the manuscript.

  1. The proliferation cannot be confirmed only using MTT assay.
  2. The author deduced that DCA Induces Intrinsic Apoptosis via the Mitochondrial Pathway, but there have only some detected quantitative indicators of cells after compound and photodynamic therapy, there should design more precise experimental to confirm this conclusion.
  3. The same drawbacks also exist in the conclusion of the ferroptosis pathway, there should at least have the detection of the pathways after using inhibitors or activators even through no related genes knock-out or knock-down analysis results.

Round 2

Reviewer 1 Report

Comments and Suggestions for Authors

I would like to comment on the authors for their thorough revisions. They have satisfactorily addressed all of my previous concerns, and the manuscript is significantly strengthened as a result.

The inclusion of the Bliss Independence analysis to formally quantify synergy, the correction of the PDT parameters, the addition of error bars and statistical analysis to the flow cytometry figures, and the responsible moderation of the claims and title all represent substantial improvements. I am now satisfied with the manuscript.

I have one final, minor suggestion for the authors' consideration regarding the statistical analysis. This is not intended as a request for another round of revision, as I do not believe in multiple rounds of review for such points and am happy to recommend the paper for publication as-is. This is simply a suggestion for the authors' consideration, perhaps for the final editorial pass or for their future work, to enhance the statistical rigor.

1. For the bar graphs with multiple groups (e.g., Figures 2, 7, 8, 9), the authors state they used a student's t-test to compare treatments to the control. A t-test is best suited for comparing two groups. When comparing three or more groups (e.g., Control, DCA, PDT, DCA+PDT), using a one-way ANOVA followed by a post-hoc test (like Tukey's or Dunnett's) is the more robust method as it avoids inflating the Type I error rate.

More importantly, an ANOVA with post-hoc comparisons would allow the authors to make the most compelling statistical comparison: whether the combination treatment (e.g., ALA+light+DCA) is significantly more effective than the mono-therapies (ALA+light and DCA alone), not just versus the untreated control. This comparison is at the heart of the paper's central claim.

2. For the time-course data in Figure 1c, the authors' note about omitting p-values for clarity is understood. The ideal statistical approach for this data structure (with factors of "Time" and "Treatment") would be a two-way ANOVA, which could formally analyze the interaction and justify post-hoc comparisons at each time point.

Again, these statistical points are offered as constructive suggestions for the authors. The paper is much improved, and the core findings are now much better supported by the data.

Reviewer 2 Report

Comments and Suggestions for Authors

I think the manuscript can be accepted in present form